# Normal-Weight Obesity and Metabolic Syndrome in Korean Adults: A Population-Based Cross-Sectional Study

**DOI:** 10.3390/healthcare11162303

**Published:** 2023-08-15

**Authors:** Jeonghyeon Kim, Seamon Kang, Hyunsik Kang

**Affiliations:** College of Sport Science, Sungkyunkwan University, Suwon 16419, Republic of Korea; zzagkim115@naver.com (J.K.); abtkang2@gmail.com (S.K.)

**Keywords:** body mass index, normal-weight obesity, metabolic syndrome, fat mass, lean mass

## Abstract

Background: The disadvantage of using body mass index (BMI) as an obesity diagnostic tool is that it cannot distinguish between fat mass and lean mass, which may understate the impact of obesity on metabolic complications. This population-based cross-sectional study aimed to investigate the relationship between normal-weight obesity (NWO) and metabolic syndrome in Korean adults aged 20 years (5962 males and 6558 females) who took part in the Korea National Health and Nutrition Examination Surveys from 2008 to 2011. Methods: NWO was defined as having a BMI of 18.5 to 24.9 kg/m^2^ and a body fat percentage of 26.0% in males or 36.0% in females. Metabolic syndrome (MetS) was defined using the revised National Cholesterol Education Program definition, with a Korean-specific waist circumference threshold of >90 cm for males and >85 cm for females. Results: NWO males and females were 2.7 times (*p* < 0.001) and 1.9 times (*p* < 0.001) more likely to develop metabolic syndrome than normal-weight non-obese males and females, respectively. Additionally, NWO females were 1.3 times (*p* = 0.027) more likely to develop MetS even after adjustments for all measured covariates. Conclusions: The current findings of the study show that NWO is a proxy biomarker of MetS to be considered for early intervention.

## 1. Introduction

Obesity, an abnormal accumulation of body fat, is a multifactorial health condition pathologically linked to metabolic disorders such as type 2 diabetes and cardiovascular disease [1]. The selection of a proper tool for the early detection of obesity is critical and clinically important for preventing and treating obesity-related metabolic complications. Traditionally, body mass index (BMI) has been used to diagnose obesity because of its advantages, such as simplicity, cost-effectiveness, and non-invasiveness. BMI is solely based on height and weight, allowing individuals to have their BMI checked with reasonable accuracy regularly [2]. BMI cutoff points based on age, gender, and ethnicity have been developed to assess obesity and associated metabolic complications [3,4]. However, BMI is an inaccurate surrogate biomarker of body fat mass because it does not consider key components of body composition, such as fat mass and fat-free mass, as well as racial and gender differences in body composition, and thus may not accurately predict health conditions [5]. A growing body of evidence suggests that considering both fat mass (FM) and lean mass (LM) is critical in determining the metabolic complications associated with overweight and/or obesity [6], particularly in Asian populations where BMI may underestimate the prevalence and severity of obesity.

Normal-weight obesity (NWO) refers to a unique body composition phenotype of excessive body fat despite having a normal BMI [5]. According to previous studies, NWO is common among Asian populations [7], but limited data are available about its clinical implications [8]. Given the limitations of BMI, therefore, attention has been given to NWO as a novel biomarker of metabolic disorders associated with overweight and obesity [9]. Recent studies showed that people with NWO have elevated metabolic risk profiles such as low-grade inflammation, impaired glucose tolerance, oxidative stress, and dyslipidemia compared with those with normal weight non-obesity [5,10,11]. Additionally, other studies suggested that NWO was pathologically linked to sarcopenia, an unfavorable atherogenic lipid profile, and insulin resistance [12]. Taken together, the findings of the previous studies imply that NWO has an impact on various morbidity and mortality risk.

In Korea, the prevalence of metabolic syndrome (MetS) ranges from 5.2 to 35.3% in males and 9.0 to 39.2% in females. Several lifestyle behaviors are involved in the etiology of MetS [3]. Obesity can appear in various ways, ranging from seemingly lean individuals to those suffering from morbid obesity. Several factors contribute to large variations in BMI among people of the same body weight, including age-related sarcopenia [13], ethnicity-specific body shape and composition [14], physical activity [14], somatotype [15], and others [16]. In this regard, NWO is a new type of obesity that should be considered in terms of metabolic issues and an under-recognized health condition that is especially prevalent in Asian populations [9]. Despite having a lower proportion of obese people, Asians have a higher prevalence of metabolic disorders than Caucasians [17,18]. Excess body fat in otherwise thin Asian phenotypes with a normal BMI may explain this disparity [9]. As a result, we hypothesized that NWO is an additional biomarker of metabolic complications associated with obesity. This study aimed to explore the association between NWO and MetS in a representative sample of Korean adults.

## 2. Materials and Methods

### 2.1. Data Source

Korea National Health and Nutrition Examination Survey (KNHANES) is a population-based cross-sectional survey designed to assess health-related behavior, health condition, and nutritional status of non-institutionalized Korean citizens living in Korea. A multi-stage clustered probability design was used for sampling. All statistics were computed using sample weights assigned to participants in the sample. The national public database contains detailed information on the KNHANES from 2008 to 2011 (https://www.cdc.gov/nchs/nhanes/about_nhanes.htm; accessed on 10 December 2022).

As shown in Figure 1, we selected participants aged 20 and older (*n* = 18,309) from the 2008–2011 KNHANES, where dual x-ray absorptiometry (DXA) for body composition assessment was included as an additional parameter. The following participants were excluded: those lacking DXA data or body composition data (*n* = 583); who were underweight (*n* = 696); and those with arthritis (*n* = 2574), liver cirrhosis (*n* = 21), chronic renal disease (*n* = 32), diabetes (*n* = 852), thyroid disease (*n* = 210), or any type of cancer (*n* = 112); and/or those with no data on serum vitamin D and/or covariates (*n* = 709). The remaining 12,520 participants (5962 men and 6558 women) were included in the final analyses.

### 2.2. Variables

#### 2.2.1. Measurement of Body Composition

Body composition was determined using whole-body DXA (Hologic, Bedford, MA, USA). Trained technologists performed standardized daily quality control on the DXA instrument using a spine phantom provided by the manufacturer. Total and trunk LM (g) and total and trunk FM (g) were measured, and the fat-to-lean mass ratio (FMR) was calculated as the whole-body FM divided by the whole-body LM. BMI and DXA-based body fat cutoffs [19]:

#### 2.2.2. Definition of Normal-Weight Obesity

Body weight and height were measured to the nearest 0.1 kg and 0.1 cm. Body mass index (BMI in kg/m^2^) was calculated by dividing body weight (kg) by height (m^2^). Normal-weight obesity (NWO) was defined as a BMI of 18.5 to 24.9 kg/m^2^ and body fat of ≥26.0% in men or ≥36.0% in women, as previously suggested for Korean populations [19]. Normal-weight non-obesity (NWNO) was defined as a BMI of 18.5 to 24.9 kg/m^2^ and body fat of <26.0% in men or <36.0% in women. Obesity (OB) was defined as a BMI of ≥25 kg/m^2^.

#### 2.2.3. Definition of Metabolic Syndrome

The National Cholesterol Education Program definition [20], along with the adoption of a Korean-specific waist circumference (WC) threshold [21], or the presence of three or more of the following criteria, were used to determine the presence of MetS: (1) WC of ≥90 cm in men or ≥85 cm in women, (2) triglycerides (TG) of ≥150 mg/dL or medication use, (3) low high-density lipoprotein cholesterol (HDLC) of <40 mg/dL in men and <50 mg/dL in women, (4) high resting systolic blood pressure (SBP) of ≥130 mmHg and diastolic blood pressure (DBP) of ≥85 mmHg or use of antihypertensive agents, and/or (5) fasting blood glucose (FBG) of ≥100 mg/dL or use of anti-diabetic medication.

#### 2.2.4. Covariates

Age (years); gender (men vs. women); education (elementary/less, middle/high school, or college/better); monthly income (in Korean won); smoking (past/current smoker vs. non-smoker); heavy alcohol consumption (7 drinks per week for males or five drinks per week for females); and nutrient intake of carbohydrates, fats, and proteins were all covariates in the study. A face-to-face interview with the 24-h recall method assessed the nutritional intake of carbohydrates, fats, and proteins. Trained personnel interviewed study participants to collect demographic data and information on their health behaviors. Additionally, the concentrations of 25-hydroxyvitamin D (ng/mL) in fasting blood samples were determined with a gamma counter (1470 Wizard, Perkin-Elmer, Turku, Finland) and a radioimmunoassay kit (DiaSorin, Stillwater, MN, USA). The Korean version of the International Physical Activity Questionnaire (https://sites.google.com/view/ipaq, (accessed on 10 December 2022)) was used to track walking and moderate and vigorous physical activity in terms of duration (min per session) and frequency (days per week), with activity measured in minutes per week and classified as inactive/insufficiently active (<150 min per week) or sufficiently active (≥150 min per week) according to WHO guidelines for physical activity [22]. Detailed procedures to assess the covariates are available elsewhere [23].

### 2.3. Statistical Analyses

The descriptive statistics of the study participants were presented as means and standard deviations for continuous variables and the number and percentages for categorical variables. The normality of the data distribution was checked before statistical analyses. ANOVA was used to test mean group differences in continuous variables (mean and standard deviation), followed by Tukey’s post hoc test if necessary. The chi-square test was used to examine discontinuous variables (in terms of number and percentage). The incremental linear trends in the prevalence of each MetS component across body composition phenotypes (from NWNO to NWO and OB) were tested using one-way ANOVA with a post-hoc contrast. The odds ratio (OR) and 95% confidence interval (CI) for metabolic syndrome were calculated using multivariate logistic regression before (crude OR) and after (adjusted OR) adjustments for all measured covariates. Statistical analyses were conducted using the SPSS-PC statistical software package (v. 27.0; IBM Corp, Armonk, NY, USA). Statistical significance was determined at *p* = 0.05.

## 3. Results

Table 1 represents the descriptive statistics of measured parameters according to body composition phenotypes. People with NWO were older and more likely to be women, married, and physically active; smoked and drank less heavily; had lower total body LM; and consumed fewer calories and proteins but had higher BMI, percent body fat, and total body FM than people with NWNO (*p* < 0.001 for all). People with NWO had higher FBG, TC, and TG levels but lower total body LM (HDLC) and serum vitamin D levels than those with NWNO (*p* < 0.001 for all). People with OB had higher SBP and DBP, as well as higher FBG, total cholesterol (TC), TG, AST, and ALT levels, but lower HDLC than people with NWO (*p* < 0.001 for all). The current findings indicate that the severity of unfavorable metabolic profiles increases with body composition phenotype (from NWNO to NWO and OB in order).

The prevalence of NWO was 19.3%, with 12.3% in men and 25.0% in women (Chi-square = 43.777, df = 1, *p* < 0.001). Figure 2 depicts the FMR according to body composition phenotypes. In the total group, individuals with NWO had a significantly higher FMR (*p* < 0.001) than individuals with NWNO, with no difference between the NWO and OB groups. In men-only analysis, individuals with NWO had a significantly higher FMR than individuals with NWNO (*p* < 0.001) or OB (*p* < 0.001). In women-only analysis, individuals with NWO had a significantly higher FMR than individuals with NWNO (*p* < 0.001), but a lower FMR (*p* < 0.001) than individuals with OB.

Table 2 presents the prevalence of individual metabolic risk factors by body composition phenotype. Regardless of gender, the prevalence of each MetS risk factor increased linearly across body composition phenotype (from NWNO to NWO and OB). People with NWO or OB had a higher prevalence of hyperglycemia, hypertriglyceridemia, hypertension, and decreased HDLC than people with NWNO (*p* < 0.001 for all). In turn, people with OB have a higher prevalence of hyperglycemia, hypertriglyceridemia), hypertension, and decreased HDLC than people with NWO (*p* < 0.001 for all).

Table 3 represents the estimated risk of MetS according to body composition phenotype. Regardless of gender, people with NWO are at an increased risk of MetS; NWO males and females had 2.7 times (*p* < 0.001) and 1.9 times *(p* < 0.001) higher risk of metabolic syndrome compared to NWNO males and females, respectively. The increased risk of MetS in men (*p* < 0.001) and women (*p* < 0.001) remained significant even after controlling for all covariates in the study. When the FMR was additionally controlled, however, NWO females only had a 1.3 times higher risk (*p* = 0.027) of MetS compared to NWNO females. Finally, obese people are at an increased risk of MetS. Specifically, obese males and females had 7.0 times (*p* < 0.001) and 9.1 times (*p* < 0.001) higher risk of MetS compared to NWNO males and females, respectively. Obese males and females had 3.7 times (*p* < 0.001) and 7.5 times (*p* < 0.001) higher risk of MetS even after adjustments for all covariates compared to NWNO males and females, respectively.

## 4. Discussion

This population-based study examined the relationship between NWO and MetS in a representative sample of Korean adults. It showed that NWO is linked to a higher prevalence of each metabolic risk factor and an increased risk of MetS. Particularly, we found that people with NWO have a distinct feature of higher FMR reflecting high total body FM and/or low total body LM, which plays an important role in determining its relationship with MetS. The current findings of the study indicate that NWO is an important risk factor in determining MetS susceptibility, though the underlying pathology is not fully understood.

Although the accuracy of BMI as a diagnostic tool for overweight and obesity has been validated in different populations, it can still underestimate the prevalence and severity of obesity in Asian populations due to smaller physiques and ethnic differences in body fat distribution [7]. Although the accuracy of BMI as a diagnostic tool for overweight and obesity has been validated in different populations, it can still underestimate the prevalence and severity of obesity in Asian populations due to relatively small physiques and ethnic differences in body fat distribution [7]. NWO has been identified as an important factor influencing health conditions, and it is linked to an increased risk of cardiometabolic disorders and premature deaths from all causes [9,24]. The cutoff values of percent body fat for the diagnosis of NWO differ depending on the study population, gender, and race, and there are no established or standardized criteria for it. [9]. Based on the NWO definition, we used the previously determined cutoff value of percent body fat at 26% for men and 36% for women who participated in this study. The prevalence of individuals with NWO in this study was lower than that of Chinese adults [25] and comparable to that of Japanese adults [26] in previous studies in which similar cutoff values were adopted. The association between NWO and cardiometabolic complications has been consistently reported in different populations [5,11,12,19].

In agreement with the current findings, evidence is mounting to establish NWO as a proxy biomarker for cardiometabolic complications [8]. Romero-Corral et al. [27] found that NWO was associated with a higher prevalence of dyslipidemia, hypertension (men), and cardiovascular disease (women), as well as an increased risk of MetS by analyzing data from ≥20-year-old 6171 subjects who participated in the Third National Health and Nutrition Examination Survey. In that study, they also showed that women with NWO were at an increased risk for cardiovascular disease mortality compared with lean women. In a cross-sectional study of 154 Iranian sedentary women aged 20–35, Ashtary-Larky et al. [28] examined the association between NWO and cardiometabolic complications. They showed that NWO was linked to an increased risk of cardiometabolic complications [19]. In another cross-sectional study involving a representative sample of 117,163 Japanese adults (82,487 men and 34,676 women) aged 40–64 years, Shirasawa et al. [26] examined the association between normal-weight central obesity and cardiovascular risk factors. They showed that normally weighted but centrally obese individuals are at an increased risk of hypertension, dyslipidemia, and diabetes. Coelho et al. [29] showed that NWO individuals had a higher incidence of MetS assessed at the ages of 37 to 39 years compared to their NWNO counterparts in an age-based cohort study involving 279 participants. Mohammadadian Khonsari et al. [30] conducted a meta-analysis involving 177,792 participants aged 13–75 years drawn from 25 previously published articles. They showed that the NWO phenotype was linked to an increased risk of dyslipidemia, hyperglycemia, diabetes, and hypertension, as well as MetS. By conducting another systematic review and meta-analysis of 19 quantitative studies, Mohammadian Khonsari et al. [31] found that people with NWO had significantly higher levels of C-reactive protein and interleukin-6 than people with NWNO. Furthermore, the negative health outcomes of NWO phenotype have been reported in non-diabetic patients with chronic kidney disease [32] and clinically stable older patients with chronic obstructive pulmonary disease [33], as well as patients with cancer [34] and coronary artery disease [35]. The findings from the current and previous studies suggest that NWO is a preclinical stage that should be routinely monitored for early intervention.

Several explanations can be given for the link between NWO and MetS. First, excessive fat accumulation in NWO syndrome may contribute to dyslipidemia, insulin resistance, decreased serum vitamin D, increased inflammatory cytokines, and oxidative stress, resulting in a clustering of risk factors for MetS [36,37]. Second, low skeletal muscle or sarcopenia in NWO syndrome might also contribute to increased pro-inflammatory cytokines, decreased anti-inflammatory cytokines, increased insulin resistance, impaired glucose metabolism, decreased serum vitamin D, decreased basal metabolic rate, and poor functional capacity and physical fitness, resulting in a clustering of risk factors for MetS [38,39]. Third, high FMR reflecting the high FM and/or low LM in NWO syndrome may be equivalent to sarcopenic obesity, associated with unfavorable cardiometabolic risk profiles, increased insulin resistance, and MetS [40,41]. On the other hand, our findings suggest that, even in individuals with high FM, maintaining LM through an active lifestyle may function to reduce susceptibility to MetS by secreting anti-inflammatory cytokines and myokines and/or suppressing proinflammatory cytokines [42], which regulate lipid and glucose metabolism [43], blood pressure [44], and MetS [45]. Fourth, unhealthy eating behaviors such as eating ultra-processed foods and drinking sweetened beverages may contribute to the etiology of the NWO phenotype [46]. It is well established that unhealthy dietary habits are associated with overweight, abdominal obesity, impairments in glucose and insulin homeostasis, abnormal lipids and lipoproteins, and elevated systemic inflammation and oxidative stress, all of which increase the risk of developing diabetes, MetS, and cardiovascular disease [47]. Fifth, a recent study [48] found that community-dwelling shift workers are prone to irregular eating times, sleep deprivation, job-related stress, and fatigue, and they are at increased risk of health conditions such as obesity, NWO syndrome, heart disease, and others by inducing hormonal changes related to circadian rhythm and appetite. However, the precise explanation(s) for the link between job conditions and NWO will be revealed in a future study. Lastly, we do not have a clear explanation for the gender disparity in NWO prevalence. As a possible explanation, the current study findings show that women have higher FMR than men regardless of body composition phenotypes (F = 316.187, df = 2, *p* < 0.001), implying that women are more likely to develop NWO associated with sarcopenia than men. Yet, the etiology of the gender difference in the prevalence of NWO phenotype and its clinical implications observed in the current study remains to be investigated in a future study.

This study has some limitations. First, because we cannot provide a cause-and-effect explanation for the current findings due to the study’s cross-sectional nature, a well-designed cohort study will be required to investigate any possible causal relationship between NWO and MetS. Second, we cannot rule out the possibility that the relationship between NWO and MetS varies depending on the obesity indicator and criteria used. Third, the relationship between some unhealthy lifestyle factors and NWO was unexpectedly skewed in the opposite direction of previous research. As a result, we cannot rule out the possibility that unexplored covariates may influence the relationship between NWO and MetS.

Despite these limitations, this is the first study to investigate the relationship between NWO and MetS in a large-scale representative dataset of Korean adults with available whole-body DXA-based body composition data. As a result, the study findings are clinically relevant to strategies for reducing the global increase in MetS morbidity and mortality.

## 5. Conclusions

In conclusion, this population-based cross-sectional study investigated the relationship between NWO and MetS in a representative sample of Korean adults aged 20 years and older, establishing NWO as a proxy biomarker of MetS. Given the limitations of BMI as a diagnostic tool for overweight and obese patients, the findings indicate that people with NWO should be monitored in primary care settings for early intervention.

## Figures and Tables

**Figure 1 healthcare-11-02303-f001:**
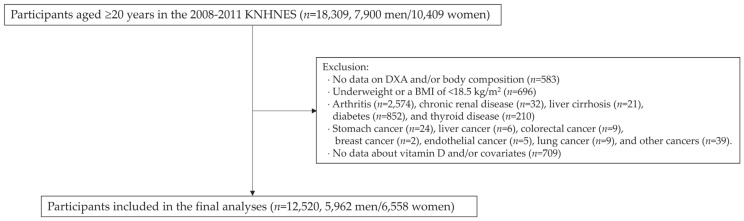
Selection of study participants. KNHNES: Korea National Health and Nutrition Examination Survey; DXA: dual x-ray absorptiometry.

**Figure 2 healthcare-11-02303-f002:**
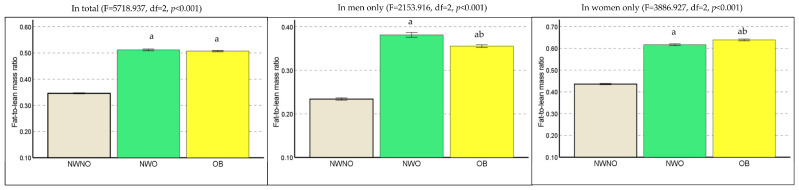
Comparison of fat-to-lean mass ratio according to body composition phenotypes. NWNO: normal-weight non-obesity; NWO: normal-weight obesity; OB: obesity. a: significantly different compared to NWNO. b: significantly different compared to OB.

**Table 1 healthcare-11-02303-t001:** Descriptive statistics of measured parameters according to body composition phenotype.

	NWNO (*n* = 6365)	NWO (*n* = 2375)	OB (*n* = 3780)	*p*-Value ^1^
Age (years)	44.5 ± 15.0	46.7 ± 15.4 ^a^	45.8 ± 13.4 ^ab^	<0.001
Sex, *n* (%)				<0.001
Men	3068 (48.2)	733 (30.9) ^a^	2161 (57.2) ^ab^	
Women	3297 (51.8)	1642 (69.1) ^a^	1619 (42.8) ^ab^	
Education, *n* (%)				0081
Elementary or lower	1034 (16.4)	413 (17.5)	675 (18.0)	
Middle/high school	3141 (49.7)	1112 (47.2)	1839 (49.0)	
College or higher	2142 (33.9)	831 (35.3)	1240 (33.0)	
Marital status, *n* (%)				<0.001
Married	5174 (81.3)	2017 (84.9) ^a^	3292 (87.1) ^ab^	
Never married	1181 (18.6)	354 (14.9) ^a^	479 (12.7) ^ab^	
Other	10 (0.2)	4 (0.2)	9 (0.2)	
Current/past smoker	1169 (26.4)	376 (16.0) ^a^	1049 (27.9) ^b^	<0.001
Heavy drinking, *n* (%)	1026 (16.2)	279 (11.8) ^a^	778 (20.7) ^ab^	<0.001
Physical activity status				<0.001
Inactive/insufficient, *n* (%)	2636 (41.6)	823 (34.9) ^a^	1628 (43.2) ^ab^	
Sufficient, *n* (%)	3705 (58.4)	1538 (65.1) ^a^	2138 (56.8) ^ab^	
Body composition				
BMI (kg/m^2^)	21.8 ± 171	22.9 ± 1.5 ^a^	27.4 ± 2.2 ^ab^	<0.001
Body fat (%)	24.1 ± 6.3	33.6 ± 4.9 ^a^	30.5 ± 7.3 ^ab^	<0.001
WC (cm)	76.0 ± 6.8	79.4 ± 6.8 ^a^	90.1 ± 7.2 ^ab^	<0.001
FM (g)	13,809 ± 3342	19,684 ± 2659 ^a^	22,248 ± 5390 ^ab^	<0.001
LM (g)	44,403 ± 8465	39,543 ± 7253 ^a^	51,412 ± 9999 ^ab^	<0.001
FMR	0.33 ± 0.11	0.51 ± 0.11 ^a^	0.46 ± 0.16 ^ab^	<0.001
Nutrient intake				
Caloric intake (kcal/day)	2382 ± 948	1681 ± 636	1995 ± 864	<0.001
Carbohydrates (% energy)	67.6 ± 10.8	67.7 ± 10.9	67.3 ± 11.4	0.302
Fats (% energy)	17.8 ± 8.7	17.7 ± 8.7	17.8 ± 8.9	0.792
Proteins (% energy)	14.7 ± 4.2	14.6 ± 4.1 ^a^	14.9 ± 4.5 ^a^	0.010
Metabolic risk factors				
Systolic BP (mmHg)	115.4 ± 16.7	116.2 ± 16.8	122.4 ± 16.1 ^ab^	<0.001
Diastolic BP (mmHg)	75.2 ± 10.7	75.3 ± 10.3	80.9 ± 10.9 ^ab^	<0.001
FBG (mg/dL)	92.3 ± 13.9	93.5 ± 12.1 ^a^	98.5 ± 18.3 ^ab^	<0.001
TC (mg/dL)	181.4 ± 33.4	191.2 ± 35.1 ^a^	196.8 ± 36.1 ^ab^	<0.001
TG (mg/dL)	114.6 ± 103.0	127.4 ± 92.4 ^a^	167.9 ± 130.4 ^ab^	<0.001
HDLC (mg/dL)	50.8 ± 11.6	48.9 ± 10.9 ^a^	44.8 ± 9.7 ^ab^	<0.001
ALT (IU/L)	21.5 ± 14.6	20.8 ± 9.2	24.6 ± 14.2 ^ab^	<0.001
AST (IU/L)	18.8 ± 18.0	19.4 ± 12.9	29.0 ± 25.3 ^ab^	<0.001
Vitamin D (ng/mL)	18.5 ± 7.0	17.1 ± 6.3 ^a^	18.6 ± 6.4 ^b^	<0.001

NWNO, normal-weight non-obesity; NWO, normal-weight obesity; BMI, body mass index; FM, fat mass; LM, lean mass; OB, obesity; FMR, fat-to-lean mass ratio; BP, blood pressure; FBG, fasting blood glucose; TC: total cholesterol; TG: triglycerides; HDLC, high-density lipoprotein cholesterol; ALT, alanine aminotransferase; AST, aspartate transaminase. Heavy drinking was defined as seven drinks per day for men and five drinks per day for women. ^1^ One-way ANOVA was used to compare continuous data among the three groups, and the post-hoc analysis was evaluated using the Tukey test. The comparison of categorical variables among groups was performed using a Chi-square test. ^a^ Significant difference between NWNO and NWO or NWNO and OB at *p* < 0.05. ^b^ Significant difference between NWO and OB at *p* < 0.05.

**Table 2 healthcare-11-02303-t002:** Prevalence of metabolic syndrome risk factors according to body composition phenotype.

	NWNO	NWO	OB	*p*-Value
Total (*n* = 12,520)				
Hyperglycemia	1059 (16.7)	466 (19.7) ^a^	1265 (33.5) ^ab^	<0.001
Hypertriglyceridemia	1459 (22.9)	741 (31.2) ^a^	1872 (49.5) ^ab^	<0.001
Hypertension	1773 (27.9)	778 (32.8) ^a^	1847 (48.9) ^ab^	<0.001
Decreased HDLC	2163 (34.0)	1081 (45.5) ^a^	1962 (51.9) ^ab^	<0.001
Men (*n* = 5962)				
Hyperglycemia	682 (22.3)	223 (30.4) ^a^	789 (36.5) ^ab^	<0.001
Hypertriglyceridemia	961 (31.3)	355 (48.4) ^a^	1247 (57.7) ^ab^	<0.001
Hypertension	1131 (36.9)	352 (48.0) ^a^	1179 (54.6) ^ab^	<0.001
Decreased HDLC	742 (24.2)	272 (37.1) ^a^	936 (43.3) ^ab^	<0.001
Women (*n* = 6558)				
Hyperglycemia	377 (11.5)	243 (14.9) ^a^	476 (29.4) ^ab^	<0.001
Hypertriglyceridemia	498 (15.1)	386 (23.5) ^a^	625 (38.6) ^ab^	<0.001
Hypertension	642 (19.5)	426 (25.9) ^a^	668 (41.3) ^ab^	<0.001
Decreased HDLC	1421 (43.1)	809 (49.3) ^a^	1026 (63.4) ^ab^	<0.001

NWNO, normal-weight non-obesity; NWO, normal-weight obesity; OB, obesity. Hyperglycemia, fasting blood glucose >100 mg/dL or drug treatment for impaired fasting glucose. Hypertriglyceridemia, triglycerides ≥150 mg/dL or drug treatment for high serum triglycerides. Hypertension, systolic blood pressure ≥130 mmHg and/or diastolic blood pressure ≥85 mmHg or drug treatment for hypertension. Decreased high-density lipoprotein cholesterol (HDLC), high-density lipoprotein cholesterol <40 mg/dL for men and <50 mg/dL for women. The comparison of categorical variables among groups was performed using a Chi-square test. ^a^ Significant difference between NWNO and NWO or NWNO and OB at *p* < 0.05. ^b^ Significant difference between NWO and OB at *p* < 0.05.

**Table 3 healthcare-11-02303-t003:** Logistic regression of metabolic syndrome according to body composition phenotype.

	Crude OR (95% CI)	*p*-Value	Adjusted OR ^1^ (95% CI)	*p*-Value	Adjusted OR ^2^ (95% CI)	*p*-Value
Total (*n* = 12,520)						
NWNO	1		1		1	
NWO	1.951 (1.716~2.221)	<0.001	2.069 (1.779~2.406)	<0.001	1.271 (1.063~1.520)	0.009
OB	7.996 (7.225~8.849)	<0.001	9.815 (8.692~11.083)	<0.001	6.092 (5.233~7.091)	<0.001
Men (*n* = 5962)						
NWNO	1		1		1	
NWO	2.688 (2.224~3.248)	<0.001	2.435 (1.952~3.038)	<0.001	1.040 (0.799~1.355)	0.941
OB	7.031 (6.152~8.036)	<0.001	8.725 (7.427~10.249)	<0.001	4.478 (3.687~5.440)	<0.001
Women (*n* = 6558)						
NWNO	1		1		1	
NWO	1.855 (1.544~2.229)	<0.001	1.780 (1.442~2.197)	<0.001	1.340 (1.035~1.734)	0.027
OB	9.051 (7.738~10.587)	<0.001	10.200 (8.464~12.293)	<0.001	7.452 (5.818~9.545)	<0.001

OR, odds ratio; CI, confidence interval; NWNO, normal-weight non-obesity; NWO, normal-weight obesity; OB, obesity. OR ^1^, adjusted for age; sex; marital status; education; heavy alcohol consumption; smoking; vitamin D; physical activity; and daily intake of calories, carbohydrates, fats, and proteins. OR ^2^, adjusted for all covariates for OR ^1^ plus fat-to-lean mass ratio (FMR). Metabolic syndrome is three or more metabolic risk factors, including abdominal obesity, hyperglycemia, hypertriglyceridemia, hypertension, and decreased high-density lipoprotein cholesterol.

## Data Availability

Access to data is permitted by the Korea Centers for Disease Control and Prevention (KCDC), and requests to access data may be submitted to Hyunsik Kang (hkang@skku.edu).

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
