# Peer review of "Normal-Weight Obesity and Metabolic Syndrome in Korean Adults: A Population-Based Cross-Sectional Study"

_healthcare, 2023, doi:10.3390/healthcare11162303_

Round 1
Reviewer 1 Report
It has been a pleasure to read the article “Association between Normal-Weight Obesity and Metabolic Syndrome in Korean Adults: A Population-based Cross-sectional Study”. One of the strengths of this study is that is the first study to investigate the impact of normal-weight obesity on metabolic syndrome in a large-scale representative dataset of Korean adults with available whole-body DXA-based body composition data.
Please find some comments and suggestions on how to improve the quality of the manuscript.
In Table 1, have you tried ANCOVAS including covariables, for example, sex, in the dietary, metabolic, and body composition variables? We know that there are sex differences in body composition and metabolic variables (e.g., HDL has different cut-off points for men and women.
In table 1. Regarding the intake of macronutrients, instead of including grams per day, they could include the percentage of energy from each macronutrient concerning total energy, it would be more useful, significantly since they do not segregate by sex, it is logical that the amounts in g/day are lower in the NWO group.
Discussion:
Could you explain the sex differences in the presence of NWO and discuss what implications this difference may have? Perhaps there is a greater underdiagnosis of the risk of Metabolic Syndrome/CVD in females when using BMI alone as a risk factor for cardiovascular risk and metabolic syndrome.
Perhaps the higher number of women with NWO is the cause of lower caloric and macronutrient intake in this group? Maybe this is the reason for this finding: “The direction of the relationship between unhealthy lifestyle choices and NWO was unexpectedly the opposite of that found in previous studies”.
Author Response
In our Responses to the Comments and Suggestions by Reviewer #1
We deeply appreciate the reviewers for their thoughtful comments. We did our best to address all the comments/critics point-by-point, which are highlighted in yellow color.
Q1) In Table 1, have you tried ANCOVAS including covariables, for example, sex, in the dietary, metabolic, and body composition variables? We know that there are sex differences in body composition and metabolic variables (e.g., HDL has different cut-off points for men and women).
ANS1) Table 1 represents the descriptive statistics of study participants by gender. We believe that ANCOVAs are not necessary for that purpose. In Table 3, however, the cut-off values of waist circumference and HDLC are gender specific. The Odd ratios are adjusted for all the covariates included in the study.
Q2) In Table 1. Regarding the intake of macronutrients, instead of including grams per day, they could include the percentage of energy from each macronutrient concerning total energy, it would be more useful, significantly since they do not segregate by sex, it is logical that the amounts in g/day are lower in the NWO group.
ANS2) Thanks for the suggestions. As recommended, dietary intake of macronutrients is expressed as the percentage of total energy intake (please refer to Tables 1 and 2).
Discussion:
Q3) Could you explain the sex differences in the presence of NWO and discuss what implications this difference may have? Perhaps there is a greater underdiagnosis of the risk of Metabolic Syndrome/CVD in females when using BMI alone as a risk factor for cardiovascular risk and metabolic syndrome. Perhaps the higher number of women with NWO is the cause of lower caloric and macronutrient intake in this group? Maybe this is the reason for this finding: “The direction of the relationship between unhealthy lifestyle choices and NWO was unexpectedly the opposite of that found in previous studies”.
ANS3) Thanks for the comments about the gender difference in the presence of NWO. The following explanation is added to the Discussion as follows: “Lastly, we do not have a clear explanation for the gender disparity in NWO prevalence. As a possible explanation, the current study findings show that women have higher FMR than men regardless of body composition phenotypes (F=316.187, df=2, p<0.001), implying that women are more likely to develop NWO associated with sarcopenia than men. Yet, the etiology of the gender difference in the prevalence of NWO phenotype and its clinical implications observed in the current study remains to be investigated in a future study.”

Reviewer 2 Report
Dear authors
thank you for your submission
the topic was interesting and easy to follow
we have few suggestions to improve the impact of the manuscript
1. ABSTRACT
study rational, the aim was missing ?
no description of study type nor no. of participants
nor the parameters taken?
conclusion better to be revised in more concise way
INTRODUCTION
the gap in knowledge was not highlighted
study drive or rationale was missing
METHODS
well described
was there ethical approval granted by an institute?
RESULTS
figures; we suggest adding bold colors to highlight the differences in bar charts
DISCUSSION
the difference in female-to-male risk, which was still present for females and absent for males, was not discussed.
this is an important study finding that was missed in the discussion and recommendation; revised, please
study implication
on the current knowledge and how it will affect our daily practice is missing
REFERENCES
kindly update old references
no comments
Author Response
In our Responses to the Comments and Suggestions by Reviewer #2
We deeply appreciate the reviewers for their thoughtful comments. We did our best to address all the comments/critics point-by-point, which are highlighted in yellow color.
- ABSTRACT
Q1) study rational, the aim was missing ?, no description of study type nor no. of participants, nor the parameters taken?, and conclusion better to be revised in more concise way
ANS1) Thanks for the comments about the Abstract. The abstract including rationale, aim, design, and number of study participants, is revised entirely (refer to the Abstract). Please understand that all the parameters included in the study were not mentioned in the Abstract because of a 200-word limit.
INTRODUCTION
Q2) the gap in knowledge was not highlighted and study drive or rationale was missing
ANS2) Thanks for the comment. The knowledge gap and study rationale are clearly stated as follows:
“BMI, on the other hand, is an inaccurate measure of body fat content because it does not account for muscle mass, bone density, overall body composition, or racial and gender differences, and it cannot accurately predict the health conditions of different demographics and races [5]. A growing body of evidence suggests that considering both fat mass (FM) and lean mass (LM) is critical in determining the metabolic complications associated with overweight and/or obesity [6], particularly in Asian populations where BMI may underestimate the prevalence and severity of obesity.
The term normal-weight obesity (NWO) refers to a unique body composition phenotype of excessive body fat despite having a normal BMI [5]. According to previous studies, NWO is common among Asian populations [7], but limited data are available about its clinical implications [8]. Given the limitations of BMI, therefore, attention has been given to NWO as a novel biomarker of metabolic disorders associated with overweight and obesity [9].”
METHODS
Q3) was there ethical approval granted by an institute?
ANS3) According to the journal guidelines, this is included in the IRB statement at the end of the manuscript as follows: “Institutional Review Board Statement: The Institutional Review Board of the Korean Institute for Health and Social Affairs reviewed and approved this study (approval no. 2008-04EXP-01-C, 2009-01CON-03-2C, 2010-02CON-21-C, and 2011-02CON-06-C).”
RESULTS
Q4) figures; we suggest adding bold colors to highlight the differences in bar charts.
ANS4) Thanks. Bar charts are redrawn with different colors as follows:
DISCUSSION
Q5) the difference in female-to-male risk, which was still present for females and absent for males, was not discussed. this is an important study finding that was missed in the discussion and recommendation; revised, please
ANS5) Thanks for the comments. Due to the cross-sectional nature of the study, we cannot provide any causal explanation for the gender difference in NWO and its impact on metabolic syndrome. However, one possible explanation is now given to the Discussion as follows: “Lastly, we do not have a clear explanation for the gender disparity in NWO prevalence. As a possible explanation, the current study findings show that women have higher FMR than men regardless of body composition phenotypes (F=316.187, df=2, p<0.001), implying that women are more likely to develop NWO associated with sarcopenia than men. A further study will be necessary to investigate the etiology of the gender difference in the prevalence of NWO phenotype observed in the current study.
Q6) study implication, on the current knowledge and how it will affect our daily practice is missing
ANS6) Given the limitation of BMI as a diagnostic tool for overweight and obese patients, the current findings of the study suggest that people with NWO should be monitored in clinical settings for early intervention.
REFERENCES
Q7) kindly update old references
ANS7) Thanks. Some references (#1, #4, #10, #17) are now updated with recent ones.
- Grundy, S.M. Metabolic syndrome update. Trends Cardiovasc Med. 2016, 26, 364-373.
- Caleyachetty, R.; Barber, T.M.; Mohammed, N.I.; Cappuccio, F.P.; Hardy, R.; Mathur, R.; Banerjee, A.; Gill, P. Ethnicity-specific BMI cutoffs for obesity based on type 2 diabetes risk in England: a population-based cohort study. Lancet Diabetes Endocrinol. 2021, 9, 419-426. Erratum in: Lancet Diabetes Endocrinol. 2021, 9, e2.
- Wijayatunga, N.N.; Dhurandhar, E.J. Normal weight obesity and unaddressed cardiometabolic health risk-a narrative review. Int J Obes (Lond). 2021, 45, 2141-2155.
- Dal Canto, E.; Farukh, B.; Faconti, L. Why are there ethnic differences in cardio-metabolic risk factors and cardiovascular diseases? JRSM Cardiovasc Dis. 2018, 7, 2048004018818923.
The following references remained because they are benchmarking studies.
- Grundy, S.M.; Brewer, H.B. Jr.; Cleeman, J.I.; Smith, S.C. Jr.; Lenfant, C.; American Heart Association; National Heart, Lung, and Blood Institute. Definition of metabolic syndrome: Report of the National Heart, Lung, and Blood Institute/American Heart Association conference on scientific issues related to definition. Circulation 2004, 109, 433-438.
- Lee, S.Y.; Park, H.S.; Ki.; D.J.; Han, J.H.; Kim, S.M.; Cho, G.J.; Kim, D.Y.; Kwon, H.S.; Kim, S.R.; Lee, C.B.; Oh, S.J.; Park, C.Y.; Yoo, H.J. Appropriate waist circumference cutoff points for central obesity in Korean adults. Diabetes Res Clin Pract. 2007, 75, 72-80.

Reviewer 3 Report
Thank you for submitting to Healthcare.
What does the '†' next to an author's name mean?
Line 6: Is the email address correct?
Delete the numbers (1) (2) (3) from your abstract. Follow the guidelines of MDPI.
Line 77: same expression as line 329. so please delete it
Results: P<0.001 too many expressions. There is no need to repeat too many times. Use it only once at the end of a sentence.
Line 165: P italic?
It is not necessary to repeat the contents of Table 3 identically, including numbers. Write more concisely, focusing on representative results.
The result is overly simplistic. Write more about your main findings.
I have no specific comments.
Author Response
In our Responses to the Comments and Suggestions by Reviewer #3
We deeply appreciate the reviewers for their thoughtful comments. We did our best to address all the comments/critics point-by-point, which are highlighted in yellow color.
Q1) What does the '†' next to an author's name mean?
ANS1) The symbol is removed.
Q2) Line 6: Is the email address correct?
ANS2) Thanks. it is corrected as follows: zzagkim115@naver.com.
Q3) Delete the numbers (1) (2) (3) from your abstract. Follow the guidelines of MDPI.
ASN3) Thanks. The numbers are now removed.
Q4) Line 77: same expression as line 329. so please delete it
ANS4) Thanks. Line 329 is removed.
Q5) Results: P<0.001 too many expressions. There is no need to repeat too many times. Use it only once at the end of a sentence.
ANS5) Thanks. Statistical significance was added once at the end of a sentence as follows: “People with NWO were older and more likely to be women, married, and physically active; smoked and drank less heavily; had lower total body LM; and consumed fewer calories, carbohydrates, fats, and proteins, but had higher BMI, percent body fat, and total body FM than people with NWNO (p<0.001 for all). People with NWO had higher FBG, TC, and TG levels, but lower total body LM, HDLC), and serum vitamin D levels than people with NWNO (p<0.001 for all). People with OB had higher systolic and diastolic blood pressure, as well as higher FBG, TC, TG, AST, and ALT levels, but lower HDLC than people with NOW (p<0.001 for all).”
Q6) Line 165: P italic?
ANS6) Thanks. It is italicized: “The prevalence of NWO was 19.3%, with 12.3% in males and 25.0% in females (Chi-square=43.777, df=1, p<0.001)”.
Q7) It is not necessary to repeat the contents of Table 3 identically, including numbers. Write more concisely, focusing on representative results.
ANS7) Thanks. The description of Table 3 is entirely revised as follows: “Table 3 represents the estimated OR and 95% CI for metabolic syndrome according to body composition phenotype. Regardless of gender, people with NWO are at an increased risk of metabolic syndrome; NWO males and females had 2.7 times (p<0.001) and 1.9 times (p<0.001) higher risk of metabolic syndrome compared to NWNO males and females, respectively. The increased risk of metabolic syndrome remained statistically significant in males (p<0.001) and females (p=0.027) even after adjustments for all the covariates included in the study. When the FMR was additionally controlled, however, NWO females only had a 1.3 times higher risk (p=0.027) of metabolic syndrome compared to NWNO females. Finally, obese people are at an increased risk of metabolic syndrome. Specifically obese males and females had 7.0 times (p<0.001) and 9.1 times (p<0.001) higher risk of metabolic syndrome compared to NWNO males and females, respectively. Obese males and females had 3.7 times (p<0.001) and 7.5 times (p<0.001) higher risk of metabolic syndrome even after adjustments for all covariates compared to NWNO males and females, respectively.”

Reviewer 4 Report
I was honored to review the manuscript. The study presents high quality and deals with important clinical issues, such type of study is needed. I have only a few small remarks that the authors should address properly.
I recommend accepting the manuscript after minor revision.
There are only some points to correct:
- In the “objectives” paragraph, the aim is not clearly specified, although it is understandable when reading the whole article. Could You add one clear sentence about the intention, a problem that the article is trying to solve? Maybe a hypothesis, which will be confirmed or not in the conclusion section?
- please provide the list of abbreviations
- please provide the number of ethical approval
- - introduction and discussion section need improvement; please provide information on how your results will translate into clinical practice;
- in the discussion section please provide the study's strong points and study limitation section
- please correct typos
All the abovementioned issues are crucial for the credibility of the results. The paper can be accepted only after addressing all the issues and another subsequent review.
I recommend accepting the manuscript after minor revision.
Author Response
In our Responses to the Comments and Suggestions by Reviewer #4
We deeply appreciate the reviewers for their thoughtful comments. We did our best to address all the comments/critics point-by-point, which are highlighted in yellow color.
Q1) In the “objectives” paragraph, the aim is not clearly specified, although it is understandable when reading the whole article. Could You add one clear sentence about the intention, a problem that the article is trying to solve? Maybe a hypothesis, which will be confirmed or not in the conclusion section?
ANS1) Thanks. We provided a hypothesis and objectives as follows: “As a result, we hypothesized that NOW is an additional biomarker of metabolic complications associated with obesity. The purpose of this study was to examine the relationship between NWO and metabolic syndrome in a representative sample of Korean adults.”
Q2) please provide the list of abbreviations.
ANS2) Thanks. The list of abbreviations is not requested but provided at the end of the manuscript as follows: “Abbreviations: BMI, body mass index; FM, fat mass; LM, lean mass; FMR, fat-to-lean mass ratio; NOW, normal weight obesity; NWNO, normal weight non-obesity; OB, obesity; DXA, dual x-ray absorptiometry; KNHANES, Korea National Health and Nutrition Examination Survey; WC, waist circumference; TG, triglycerides; TC, total cholesterol; HDLC, high-density lipoprotein cholesterol; SBP, systolic blood pressure; DBP, diastolic blood pressure; FBG, fasting blood glucose; OR, odds ratio; CI, confidence interval; ANOVA, analysis of variance; ALT, alanine aminotransferase; AST, Aspartate transaminase.”
Q3) please provide the number of ethical approval.
ANS3) Thanks. The ethical approval code numbers are provided in the IRB statement section as follows: “Institutional Review Board Statement: The Institutional Review Board of the Korean Institute for Health and Social Affairs reviewed and approved this study (approval no. 2008-04EXP-01-C, 2009-01CON-03-2C, 2010-02CON-21-C, and 2011-02CON-06-C).”
Q4) introduction and discussion section need improvement; please provide information on how your results will translate into clinical practice.
ANS4) Thanks. We provided the following statement about how to translate the current study findings into clinical practice as follows: “Given the limitation of BMI as a diagnostic tool for overweight and obese patients, the current findings of the study suggest that people with NWO should be monitored in clinical settings for early intervention.”
Q5) in the discussion section please provide the study's strong points and study limitation section
ANS5) Thanks. Study limitations and strengths are provided in the Discussion as follows:
“This study has some limitations. First, the cross-sectional nature of the study limits us to provide any cause-and-effect explanation for the current findings. A cohort study will be necessary to examine any causal relationship between NWO and metabolic syndrome. Second, we cannot rule out the possibility that the relationship between NWO and metabolic syndrome varies depending on the obesity indicator and criteria used. Third, the direction of the relationship between unhealthy lifestyle choices and NWO was unexpectedly the opposite of that found in previous studies. As a result, we cannot rule out the possibility that unexplored covariates or the complex nature of the study sample limited to older adults may influence the relationship between NWO and metabolic syndrome.
Despite these limitations, this is the first study to investigate the impact of NWO on metabolic syndrome in a large-scale representative dataset of Korean adults with available whole-body DXA-based body composition data. As a result, the study findings are clinically relevant to strategies for reducing the global increase in metabolic syndrome morbidity and mortality associated with obesity.”
Q6) please correct typos.
ANS6) Thanks. Typo errors are corrected throughout the text.

Reviewer 5 Report
In the article submitted to Healthcare by Jeonghyeon Kim and colleagues entitled "Association between Normal-Weight Obesity and Metabolic Syndrome in Korean Adults: A Population-based Cross-sectional Study", the authors point to the relation between parameters in NWO individuals with the risk of metabolic syndrome. The manuscript is well introduced, the analysis is plenty described, and the results/discussion is presented clearly.
I have just minor points to raise:
- In the abstract, metabolic syndrome is not defined just by waist circumference, the authors should include the other conditions.
- In lines 34-35, the link is not working, and I think that must be a reference.
- Line 104 correct > 36% to < 36%
- What was the limit for the age? The fact of the oldest people included in the study can be a limitation, as the people present more sarcopenia than the general population.
- In Figure 1, stratified the numbers of women and men.
Author Response
In our Responses to the Comments and Suggestions by Reviewer #5
We deeply appreciate the reviewers for their thoughtful comments. We did our best to address all the comments/critics point-by-point, which are highlighted in yellow color.
Q1) In the abstract, metabolic syndrome is not defined just by waist circumference, the authors should include the other conditions.
ANS1) Thanks. Metabolic syndrome is defined in the Methods as follows; “Metabolic syndrome was defined as the presence of three or more of the following criteria: 1) waist circumference of >90 cm in men or >85 cm in women, 2) triglycerides of >150 mg/dL or medication use, 3) low high-density lipoprotein cholesterol of <40 mg/dL in men and <50 mg/dL in women, 4) high resting systolic blood pressure of >130 mmHg and diastolic blood pressure of >85 mmHg or use of antihypertensive agents, and/or 5) fasting blood glucose of >100 mg/dL or use of anti-diabetic medication.” However, the 200-word limit does not allow us to provide a full description of Metabolic Syndrome. Therefore, the definition of Metabolic Syndrome is described in brief as follows: “Metabolic syndrome was defined according to the revised National Cholesterol Education Program definition, with the adoption of a Korean-specific waist circumference threshold of >90 cm for men or >85 cm for women.”
Q2) In lines 34-35, the link is not working, and I think that must be a reference.
ANS2) Thanks. The following reference is now added: Khanna, D.; Peltzer, C.; Kahar, P.; Parmar, M.S. Body mass index (BMI): a screening tool analysis. Cureus 2022, 14, e22119.
Q3) Line 104 correct > 36% to < 36%
ANS3) Thanks. >36% is corrected to <36%.
Q4) What was the limit for the age? The fact of the oldest people included in the study can be a limitation, as the people present more sarcopenia than the general population.
ANS4) Thanks for the comments. We have 110 old-old subjects aged 81 years up to 91 years. The prevalence of NWO was found to be higher in the old-old subjects (32 out of 110 / 29.1%) than in the young-to-older adults (2343 out of 12410 / 18.9%). So, we reanalyzed the data without 110 old-old adults. Due to a relatively small sample size, however, including or excluding the old-old subjects from the data analyses did not make any significant difference in the study outcomes.
Q5) In Figure 1, stratified the numbers of women and men.
ANS5) Thanks. Stratified numbers of women and men are added to Figure 1.

Round 2
Reviewer 2 Report
Dear authors
thnk you for addressing most of our suggestions
the article was revised well
we have no further comments
no comments